# Temporal Co-Attention Guided Conditional Generative Adversarial Network for Optical Image Synthesis

**Yongchun Weng** [1,2], **Yong Ma** [1,*], **Fu Chen** [1], **Erping Shang** [1], **Wutao Yao** [1], **Shuyan Zhang** [1], **Jin Yang** [1] **and Jianbo Liu** [1]

1   Aerospace Information Research Institute, Chinese Academy of Sciences, Beijing 100094, China
2   University of Chinese Academy of Sciences, Beijing 100049, China
*   Correspondence: mayong@aircas.ac.cn

**Abstract:** In the field of SAR-to-optical image synthesis, current methods based on conditional generative adversarial networks (CGANs) have satisfying performance under simple scenarios, but the performance drops severely under complicated scenarios. Considering that SAR images can form a robust time series due to SAR's all-weather imaging ability, we take advantage of this and extract a temporal correlation from bi-temporal SAR images to guide the translation. To achieve this, we introduce a co-attention mechanism into the CGAN that learns the correlation between optically-available and optically-absent time points, selectively enhances the features of the former time point, and eventually guides the model to a better optical image synthesis on the latter time point. Additionally, we adopt a strategy to balance the weight of optical and SAR features to extract better features from the SAR input. With these strategies, the quality of synthesized images is notably improved in complicated scenarios. The synthesized images can increase the spatial and temporal resolution of optical imagery, greatly improving the availability of data for the applications of crop monitoring, change detection, and visual interpretation.

**Keywords:** attention mechanism; generative adversarial networks (GANs); image-to-image translation; synthetic aperture radar (SAR)

## 1. Introduction

According to the report of the International Satellite Cloud Climate Program (ISCCP), the annual average of global cloud coverage is as high as 66% [1]. Due to weather conditions such as cloud, fog, and haze, optical sensors often fail to obtain local ground information, resulting in particularly serious data losses, especially during the rainy season. In addition to weather conditions, sensor failures can also lead to the loss of remote sensing data. Therefore, the challenge of repairing the unattainable part of optical imagery has been a thorny problem in remote sensing.

On the hardware side, the solution to the loss of remote sensing information is to establish satellite groups and increase the observation frequency. However, increasing the observation frequency brings little benefit when image acquisition areas are frequently rainy. Sometimes there may be no available optical image for months. On the algorithm side, the commonly used methods to repair and make up for the missing information are filtering, interpolation, space spectrum fusion, and multi-temporal composition. These methods can remove small areas of thin cloud coverage pretty well, and some of them can even handle thick cloud coverage, but they are not capable of restoring large areas of missing data for a long period of time.

Therefore, neither launching new satellites nor using some traditional image processing methods can handle the problem that cloudless data cannot be obtained for a long time during the rainy season in cloudy areas. It is still very difficult to continuously monitor artificial and phenological changes in these areas.

In recent years, deep learning has been increasingly integrated with a broad range of applications of SAR imagery, including classification [2,3], target recognition [4,5], and change detection [6,7]. With the development of adversarial learning, a new possibility for integrating the spectral advantage of optical imagery and the all-weather advantage of SAR imagery has emerged. Recent advances in restoring optical imagery were largely driven by image synthesis from SAR images using GANs [8,9]. The first model that introduces the CGAN to image-to-image (I2I) translation is Pix2Pix [10], which consists of a generator G and a discriminator D. G is a U-Net architecture and D is a patch-based fully convolutional architecture. G and D compete with each other to learn the mapping from source domain items to target domain items. The objective of the generator G is to generate real items in the target domain under certain conditions, while the discriminator aims to distinguish real images from generated images. Therefore, every training sample is a pair of images $(z, x)$, where $x$ is a real image from the target domain and $z$ is a corresponding image from the source domain. The image generated under the conditions of $z$ is defined as $G(z)$. Similarly, the discrimination of image $x$ under the conditions of $z$ is defined as $D(z, x)$. Pix2Pix aims to model the conditional distribution of target domain images using the following minimax objective function:

$$L_{GAN}(G, D) = \min_{G} \left( \max_{D} \left( \mathbb{E}_{(z,x)}[\log(D(z, x))] + \mathbb{E}_{(z)}[\log(1 - D(z, G(z)))] \right) \right) \quad (1)$$

Following the success of Pix2Pix, various methods have been proposed to further boost I2I model performance, and some of them were introduced to the field of SAR-to-optical image synthesis [11–18].

Researchers' attention was first drawn to the unstable training process of GANs. To alleviate the problem, Mao et al. [19] proposed least squares generative adversarial networks (LSGANs), and Arjovsky et al. [20] proposed Wasserstein GANs (WGANs). The former employed the loss function based on the least-squares method, and the latter adapted Earth-Mover's distance to replace the Jensen-Shannon divergence, which better measured the distance between the generated and real data.

Subsequently, researchers found that as the image resolution increased, images synthesized by GANs ran into an increasingly severe problem of lacking fine details and realistic textures. This is a multi-scale problem. Different solutions have been devised to try to solve this problem. Karras et al. [21] presented StyleGANs, which use intermediate hidden variables to identify the decoupling of different levels of features. Based on StyleGANs, Richardson et al. [22] proposed an I2I translation method named PSP. Wang et al. [23] presented another solution for multi-scale generation that used a set of generators, each focusing on different levels of scale. Meanwhile, they proposed a set of discriminators to perform discrimination on different levels of scale. Multi-scale generation has become a commonly used technique since then, and it is usually combined with LSGAN or WGAN loss.

In 2019, Zhang et al. [24] introduced the self-attention mechanism to GANs. After that, a new research direction of boosting the performance of I2I translation models with attention mechanisms has been gradually catching researchers' attention. In the self-attention module, the input is first embedded into three different embedding spaces, named Q, K, and V. Then a function is applied to measure and normalize the correlation between Q and K. Usually, the dot product function is used to measure the correlation and softmax to normalize it. The correlation is regarded as a weight map, and the weighted sum of V is the output of the module. In addition, it is also common practice to add a residual connection from the input of the module to the output. Generally speaking, the attention mechanism extracts the global dependencies of features. When the attention mechanism is applied to the spatial dimension, it becomes spatial attention, which extracts spatial dependencies similar to the one used by Gao et al. [25]. When the attention mechanism is applied to the channel dimension, it becomes channel attention, which extracts channel-wise dependencies similar to the one used by Tang et al. [26]. The attention mechanism

can also be applied to the temporal dimension to infer global change information between source time and target time. This change information can guide the generator to choose different strategies for slightly and greatly changing regions and help to better learn the mapping from source time to target time.

In 2018, Bermudez and Grohnfeldt et al. [11,12] first used an adversarial learning approach to simulate optical images to compensate for information below the cloud-covered region. Ebel et al. [13] used a CycleGAN for Sentinel-2 optical image de-clouding, which reduced the requirement for data alignment. Gao et al. [14] used a GAN to generate optical images based on the fusion of pre-synthesized optical images, SAR images, and clouded optical images. They altered the input in terms of contrast and brightness to enhance the robustness of the model and reduce the spectral distortion. Zou and Li [15] improved the generated local details of Pix2Pix [10] models by adding a phase consistency constraint.

Despite many trying to adapt GAN-based I2I translation models to the field of SAR-to-optical translation, most struggle to generate satisfying results under complicated scenarios. Compared to simple scenarios, complicated scenarios contain more kinds of objects, ranging from small to medium and large scales. Various kinds of small-scale objects impose a strong challenge on modeling the transformation of the images. Taking the Pix2Pix model [10] for instance, the model derives visually much worse outputs under complicated scenarios than those under simple scenarios, as shown in Figure 1. Under the resolution of 10 m, generating satisfying results for complicated scenarios, even for some regions, is beyond the limit of Pix2Pix.

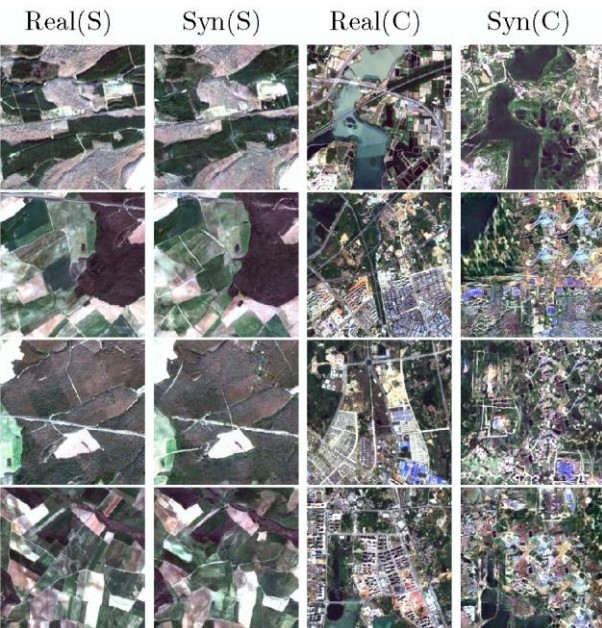

**Figure 1.** SAR-to-optical image synthesis using Pix2Pix [10] under simple and complicated scenarios. The real optical images and the images synthesized by Pix2Pix are presented in adjacent columns. The first two columns show the performance of Pix2Pix under a simple scenario, and the next two columns show its performance under a complicated scenario. The image resolution is 10 m.

To improve the generation under complicated scenarios, we intend to upgrade the GAN-based SAR-to-optical model to better face the challenge of various kinds of small-scale objects. The previous works, despite various kinds of upgrades to the model, train the generator using the source time $T_0$ (or source area) and apply it to the target time $T_1$ (or target area). However, we do not have to follow this mode because bi-temporal SAR images have good availability, and from them, the change information can be inferred to facilitate better generation. If the optical features of $T_0$ are also available, they can help with the generation of $T_1$ as well. These characteristics have not received enough attention.

Therefore, we use a temporal co-attention guided generator to extract change features from bi-temporal datasets that we have built. The change features are merged with polarization features and optical features. Apart from these features, the features of different scales are also extracted through multi-scale generation.

This paper is structured as follows: Section 2 provides detailed instructions for the bi-temporal SAR-optical dataset that we built. Then the key techniques in our method are illustrated in Section 3. Thereafter, settings of experiments and evaluation results are presented in Section 4 with some discussion of the results. Finally, the conclusion and an outlook on future work are presented in Section 5.

## 2. Dataset

The SEN1-2 dataset [27] is one of the best-known datasets for generating artificial optical images from SAR inputs. It is composed of 282,384 paired image patches, collected from a Sentinel-1 GRD product and a Sentinel-2 Level-1C product. Its top-level folders are organized according to meteorological seasons. This dataset contributes a great deal to the research regarding SAR-to-optical translation; however, samples from simple scenarios and complicated scenarios are mixed up in this dataset. Additionally, current datasets only provide single polarization or single time phase samples. In order to evaluate generative models' performance under different scenarios and incorporate polarization and change features into the generation process, we produced two new bi-temporal datasets using Sentinel-1 SLC products and Sentinel-2 Level-2A products, which include a simple scenario dataset and a complicated scenario dataset. Figures 2 and 3 show some training samples from our datasets.

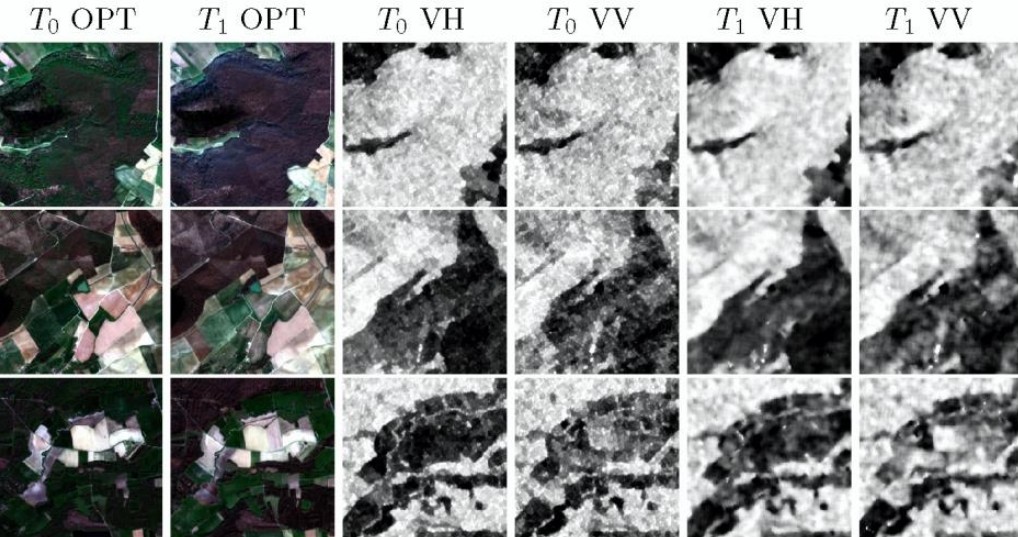

**Figure 2.** Simple scenario training sample example.

### 2.1. Dataset Procedure

We use SNAP software for Sentinel-1 and Sentinel-2 data processing. Since the Sentinel-1 SAR uses side-view imaging, it has geometric features such as foreshortening, layover, and shadowing. The geometric distortion caused by foreshortening is easily handled by applying Precise Orbit Ephemerides files and terrain corrections. In order to minimize the effect of layover and shadowing, we collect image samples in plain areas. In addition, SAR images are susceptible to noise, so we apply multi-look and Refined-Lee filtering to reduce the noise of radiation-calibrated backscatter coefficients. After all the processes mentioned above, the linear-scaled backscatter coefficient $\sigma_0$ is converted to the dB scale. Finally, all SAR images are cropped by a pre-defined grid vector to obtain the training and testing sets.

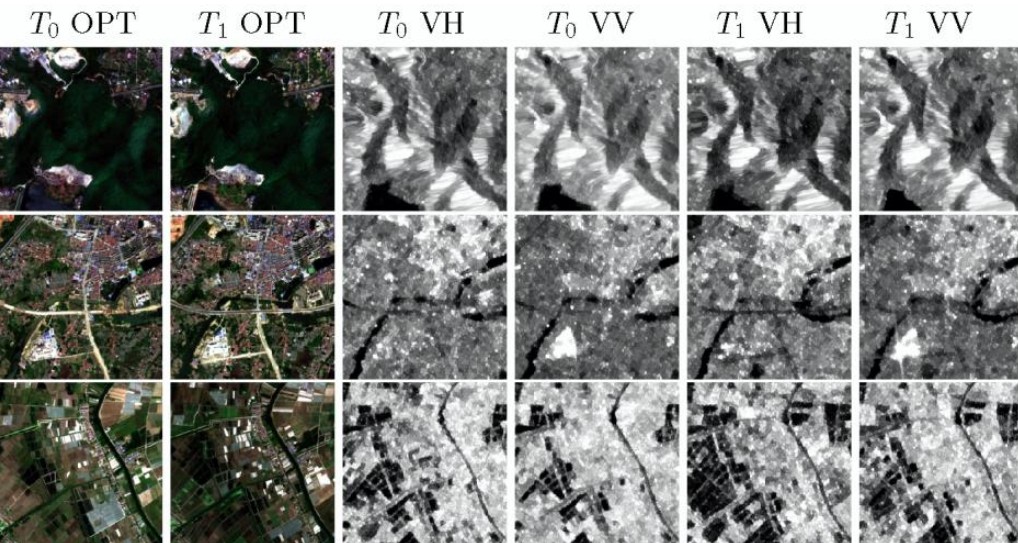

**Figure 3.** Complicated scenario training sample example.

As for the Sentinel-2 L2A Bottom-of-Atmosphere products, they have already been ortho-corrected, geometrically refined, and atmospherically corrected, so they are only required to separate the RGB bands, and then the reflectance ranging from 0 to 1 can be obtained by substituting the DN value of the L2A products into Equation (2).

$$p_{BOA} = \frac{DN - BOA\_ADD\_OFFSET}{BOA\_QUANTIFICATION\_VALUE} \tag{2}$$

The parameters $BOA\_ADD\_OFFSET$ and $BOA\_QUANTIFICATION\_VALUE$ in Equation (2) can be found in the metadata of L2A products. Finally, all optical images are also cropped into patches by a pre-defined grid vector.

### 2.2. Special Features of the Dataset

Unlike the SEN1-2 dataset, we gathered SAR-optical image pairs of the same scene from two time phases in order to evaluate how the bi-temporal SAR input and the optical input can influence the SAR-to-optical generation. So, the final training samples contain six optical channels and four SAR channels coming from two time phases with a size of $256 \times 256$ (see Figures 2 and 3).

Another difference is the division of simple and complicated scenarios. The learning difficulty of the generation varies for different scenarios. A scenario that contains more small objects means more diversity in texture and a more complex mapping from the source domain to the target domain. Therefore, we use a segmentation-based approach to evaluate the level of scale for our datasets and the SEN1-2 dataset. The method is based on the idea that, under the same scenario of acquisition time and segmentation strategy, the more segments that are derived, the smaller the scale of objects in the scenario. In this case, a commendable unsupervised segmentation method for RGB images named BASS [28] is used. We performed segmentation on the optical channels of our datasets and the SEN1-2 dataset from the same season using the same settings, and the result is shown in Figure 4. Under the same segmentation settings, the distribution of the segmented area derived from the complicated scenario is located in a lower position, while the distribution derived from the simple scenario is located in a higher position, indicating that the complicated scenario contains more small-scale objects. In contrast, the SEN1-2 dataset mixes simple and complicated scenarios, so its distribution is located in the middle with two peaks.

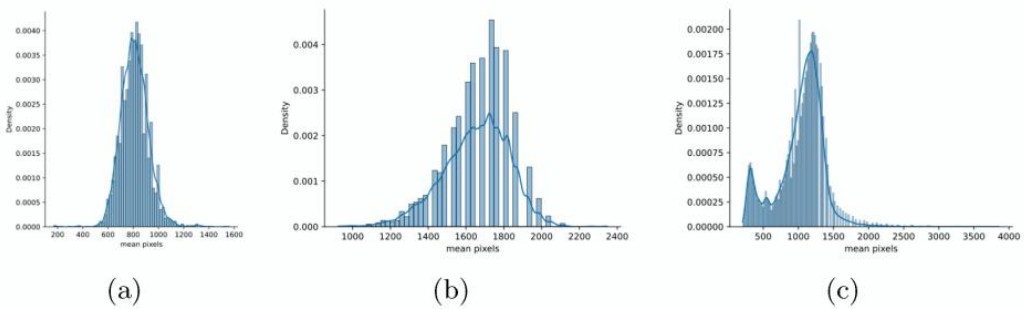

**Figure 4.** Distribution of segmented area derived from (**a**) a complicated scenario, (**b**) a simple scenario, and (**c**) the SEN1-2 dataset.

## 3. Methodology

As we have illustrated in Section 1, the commonly used I2I model Pix2Pix's capability varies largely for different scales of content. The model learns to generate better results for simple scenarios than for complicated scenarios involving various kinds of small-scale objects. We explore two ways to solve the challenge of complicated scenarios: one is by increasing model complexity to learn and overcome the challenges of mapping small-scale objects, and another is by providing the model with more information to decrease the learning difficulty. On the one hand, we provide the model with additional optical imagery and bi-temporal SAR imagery. On the other hand, we propose a temporal co-attention guided generator to extract change features from bi-temporal SAR imagery and merge them with polarization features, optical features, and multi-scale features. The model learns to generate $T_1$ optical images in a coarse-to-fine manner based on the merged features. These techniques improve the GAN's ability to synthesize optical images while keeping the model complexity acceptable. Based on the bi-temporal dataset, the training and inference procedures of our upgraded model are summarized in Figure 5. First, the satellites' data is processed into input patches and label patches. The input patches contain 3 optical bands and 4 SAR bands. Then we split these patches into a training set and a testing set. The CGAN-based model with three key techniques is trained on the training set. Finally, the inference is performed on the testing set. These steps are the same for both simple and complicated scenarios.

### 3.1. Coarse-to-Fine Generation

In order to get better features for both large-scale and small-scale content, we utilize the multi-scale generation and discrimination from Pix2PixHD.

### 3.1.1. Multi-Scale Generation

Multi-scale generation is frequently used in recent practice [23,29–35]. The main idea is that information at different scales can be aggregated to derive better performance for the image synthesis tasks. The multi-scale generator we use is composed of a global generator $G_1$ and one or more local enhancer generators $\{G_2, \cdots, G_m\}$. Every generator $G_k$ has a similar architecture: a convolutional front-end $G_k^{(F)}$, a residual block $G_k^{(R)}$, and a transposed convolutional back-end $G_k^{(B)}$. For a local enhancer generator $G_k$, its residual block takes the sum of the last feature map of $G_k^{(F)}$ and that of $G_{k-1}^{(B)}$ as the input. For instance, in the two-scale generation architecture, the last feature map of $G_2^{(F)}$ and $G_1^{(B)}$ are added together and fed to the residual block of $G_2$, which is shown in Figure 6. This is helpful for integrating larger-scale information into the local enhancer. We use two-scale generation in our method, which is enough in most cases, but additional local enhancer generators can be used to learn features on more levels of scale.

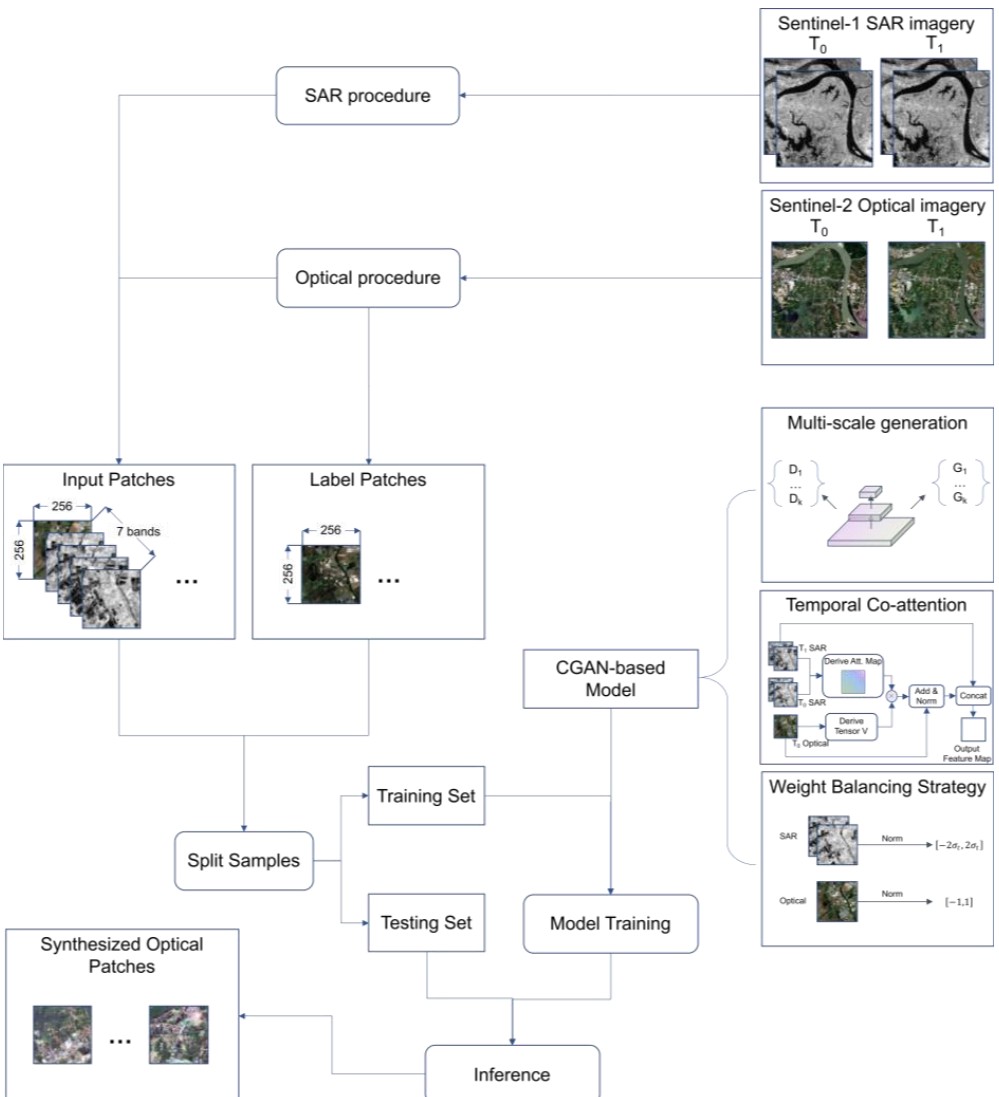

**Figure 5.** Flow chart of major steps and processes.

### 3.1.2. Multi-Scale Discrimination

Using multiple GAN discriminators has been proven effective for better guiding the generator's training [23,36,37]. The multi-scale discrimination that we use is a combination of three discriminators. As shown in Figure 7, the discriminators have an identical network structure but operate with different receptive fields. Therefore, the discrimination is performed at different levels of scale. This combination encourages the generator to yield both globally consistent and locally detail-rich output. With the multi-scale discrimination, the loss functions of the discriminators and generators are defined as Equation (4) and Equation (3), respectively:

$$L_{GAN}(G_1, G_2) = \frac{1}{2} \sum_{k=1}^{3} \mathbb{E}_{(S)} \left[ D_k \big( S, G_2 \big( S, G_1(S') \big) \big)^2 \right] \tag{3}$$

$$L_{GAN}(D) = \frac{1}{2} \sum_{k=1}^{3} \left( \mathbb{E}_{(S, X_t)} \left[ (D_k(S, X_t) - 1)^2 \right] + \mathbb{E}_{(S)} \left[ (D_k(S, G_2(S, G_1(S'))) + 1)^2 \right] \right) \tag{4}$$

These definitions are following LSGANs' practice [19] for stable training, where $X_t$ is the target optical image, and $S$ is the corresponding input of the generator that serves as the condition.

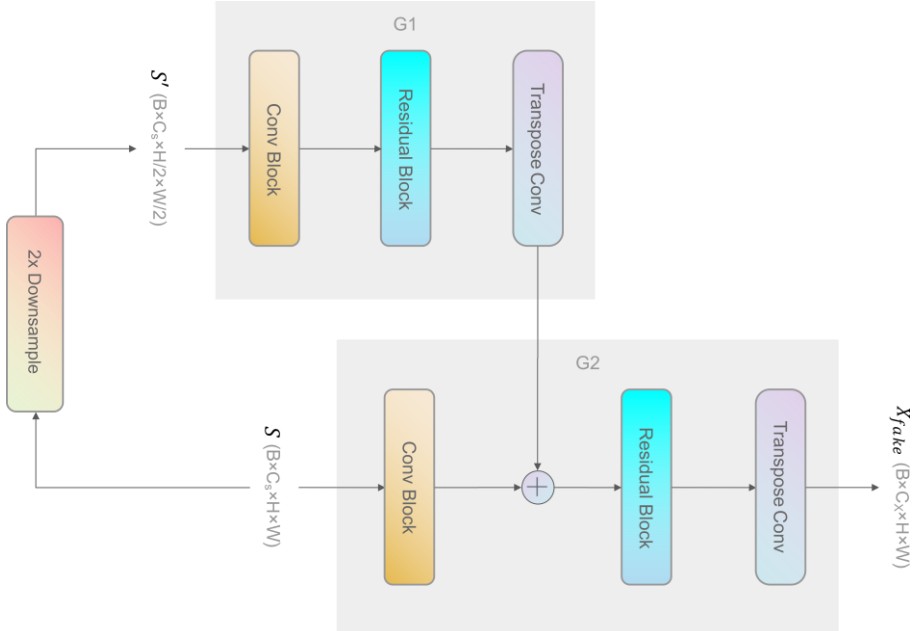

**Figure 6.** The architecture of the two-scale generation. It is composed of two generators: $G_1$ and $G_2$. $G_2$ takes the original input $S$, with the original height (H) and width (W). $G_1$ takes $S'$ as input, which is downsampled from $S$ with a factor of 2 in both the height and width (H/2, W/2).

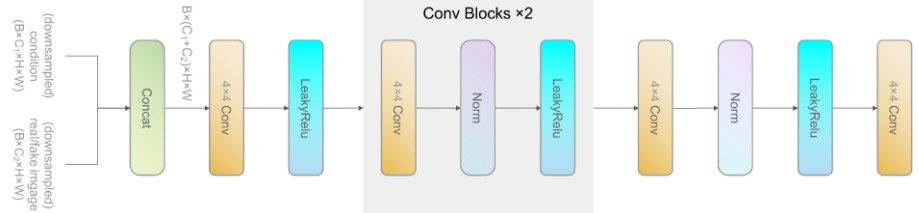

**Figure 7.** The architecture of a single discriminator. All discriminators in $\{D_1, D_2, D_3\}$ share this architecture. The input is downsampled by factors of 2 and 4, then fed to $D_2$ and $D_3$, respectively, while $D_1$ takes the original image as input.

Furthermore, the perceptual loss [38] can be added to the generator's loss function to produce a favorable result. The perceptual loss is computed by extracting the L1 loss from a pretrained VGG19 network. The final loss function of the generator is defined by Equation (5), where $\mathrm{F}^{(i)}$ is the $i$th layer with $M_i$ elements of the pretrained VGG19, and the weight of perceptual loss is controlled by $\lambda$.

$$L(G) = L_{GAN}(G_1, G_2) + \lambda \sum_{i=1}^{N} \frac{1}{M_i} \left( \| \mathrm{F}^{(i)}(X_t) - \mathrm{F}^{(i)}(G_2(S, G_1(S'))) \|_1 \right) \tag{5}$$

### 3.2. Temporal Co-Attention Guided Generator

Previous works on SAR-to-optical translation have not taken change features into consideration. Generating optical images directly from SAR images faces the challenge of great differences in sensing mechanisms and statistical distribution, so some works have tried to input the model with additional data from other times or additional raster bands [12–14,17,39], but none of them have designed a network structure to efficiently extract the change features and mix them with the optical features. Inspired by the previous success of attention mechanisms in GANs, we propose to use an attention mechanism to extract temporal dependencies of SAR images and mix them with optical features

and polarization features so as to further increase the performance of SAR-to-optical translation models.

By feeding Pix2PixHD either SAR images from target time $T_1$ or optical images from source time $T_0$, we found that generating from optical input gets visually better output but lacks awareness of regional changes compared to generating from SAR images. The reason for this is that, although the optical input is from another time, it is still more similar to the optical output in terms of texture than the SAR input. However, it does not hold any information about changes from $T_0$ to $T_1$, so the regional changes cannot be inferred. That is where SAR input can be used to implement the optical input because SAR images can form a reliable time series. By applying an elaborately designed temporal co-attention header to a generator network, we were able to train the generator to mainly rely on $T_0$ optical data while inferring regional changes with the help of the $T_0$ and $T_1$ SAR input.

In the temporal co-attention header (see Figure 8), the inputs $X$ and $Z$ are from source time $T_0$, while $Y$ is from target time $T_1$. Three $1 \times 1$ convolutional layers are applied to transform $X$, $Y$, and $Z$ into value tensor $V$, query tensor $Q$, and key tensor $K$, respectively. To limit the use of compute resources in the experiment, $V$, $Q$, and $K$ are downsampled by applying average pooling layers. Then the attention map is computed by:

$$M = Softmax\left(\frac{Q^T K}{\sqrt{C}}\right) \tag{6}$$

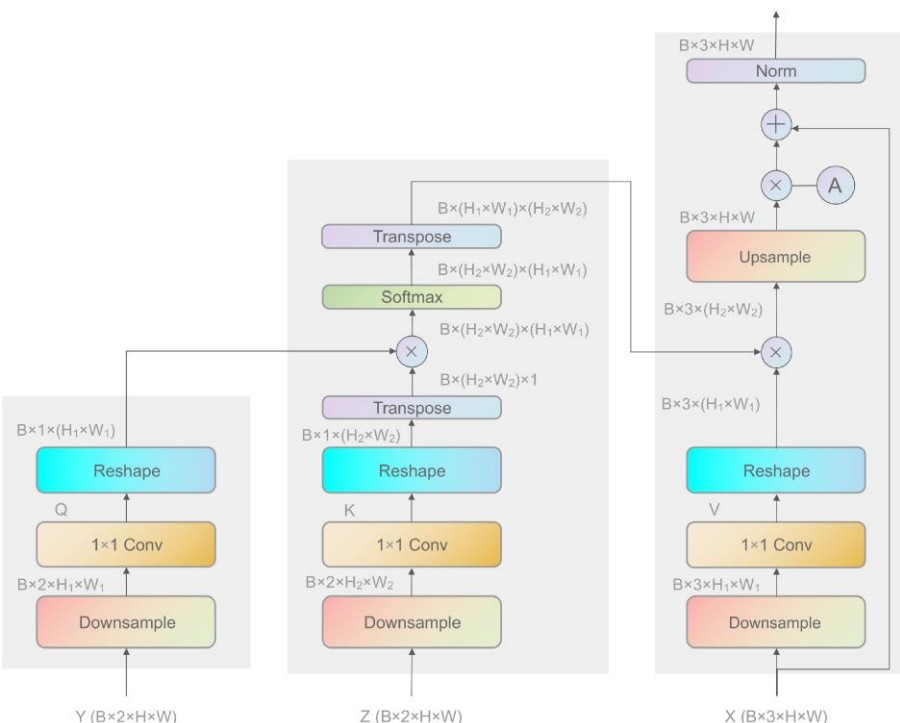

**Figure 8.** The architecture of the temporal co-attention header.

After that, following the practice of SAGAN [24], we multiply the attention output by a scale parameter $\gamma$ and add back the input $X$ (Equation (7)). Then, $Y$, $Z$, and $O$ are concatenated to form the input of the next module.

$$O = Norm(\gamma X + MV) \tag{7}$$

Now, let us analyze this process more closely. The attention map $M$ derived from Equation (6) is a measurement of the correlation between $Q$ and $K$, which are derived from $Y$ and $Z$. With $Y$ and $Z$ being set to SAR data from $T_1$ and $T_0$, respectively, the attention map can indicate the correlation between the features of $T_0$ and $T_1$. Then we set the input $X$

to optical data from $T_0$ and perform the operation defined in Equation (7) with $M$. If there is little change at a location (*i*,*j*) between $T_0$ and $T_1$, we can infer a high correlation and that optical features are reliable at that location, so the optical features at (*i*,*j*) are enhanced on the attention output; otherwise, the optical features are weakened and SAR features are taking a dominant role at that spot. Therefore, appending this module to the front of two-scale generators empowers the generators with the ability to extract and merge $T_0$ optical features, $T_1$ SAR features, change features, and multi-scale features, putting the generators under the guidance of temporal co-attention.

### 3.3. Tackling Data Heterogeneity between Optical and SAR

The optical and SAR images are both the input of the networks, but the intensity of these two kinds of data is measured very differently. To be specific, the backscatter coefficient $\sigma_0$ of SAR images can fluctuate, ranging from $-30$ to 10 dB, by just moving a few meters from buildings to roads. Regarding Sentinel-2 L2A products, the data measures bottom-of-atmosphere reflectance, which is quantified by integer values between 0 and 65,535 [40]. The integer values of BOA reflectance are converted to float values ranging from 0 to 1, as demonstrated in Section 2.1. We found that this difference in dynamic range did not cause any program breakdown, but if we do not account for it, the training process will be led in an odd direction for two reasons. On the one hand, due to the architecture of the generator, the output is restricted to the range of $(-1,1)$. In order to meet that value interval, we perform an instance normalization followed by a value clip described by Equation (8) on each optical image, with the parameters being set to $m_t = 0$ and $\sigma_t = 0.5$. After the transformation, the optical data is normalized to fall within that interval. On the other hand, normalization of optical data indicates normalization of SAR data. If we only normalize optical data, the networks will be misled into giving preference to SAR data. However, if we normalize SAR imagery to the same range as optical imagery, the output will heavily rely on optical input because of its advantages over SAR input in terms of output similarity and channel number. Therefore, we adopt a weight-balancing strategy, normalizing SAR imagery with $m_t = 0$ and $\sigma_t = 1$, to give a prior preference to SAR imagery.

$$x_{norm} = min\left(max\left(\frac{x - E[x]}{\sqrt{Var[x] + \epsilon}}\sigma_t, -2\sigma_t\right), 2\sigma_t\right) + m_t \tag{8}$$

## 4. Results and Discussion

In this section, we provide comparisons against two other methods using a simple scenario dataset. Then we show comparisons against more leading methods using a complicated scenario dataset. Finally, we report an ablation study. In all experiments, we present visual inspections and apply a number of commonly used metrics to quantify the quality of our results.

### 4.1. Experiment Settings

Our networks are implemented using Pytorch [41] and trained on GTX 3090Ti GPUs. To optimize our networks, we alternate between one gradient descent step on D and then one step on G. Some detailed parameter settings are shown in Table 1. To guarantee the impartiality of the experiments, we train all of the evaluated models with an equal number of epochs and use the official public code.

### 4.2. Baseline

We use a simple-setting Pix2PixHD as the baseline model for comparison, which means that the multi-scale discrimination is kept in the baseline model but the local enhancer network is removed. The generator of the baseline model is supplied with an input that is a channel-wise concatenation of bi-temporal SAR imagery. This baseline model provides us with a reference for performance when the generator is not improved to merge additional optical features, change features, polarization features, and multi-scale features.

The baseline model is compared with our model under simple and complicated scenarios, along with other models. Particularly, we compare the generalization performance of the baseline model with that of our model under a complicated scenario.

**Table 1.** Detailed configuration of the experiments.

| Parameters | Values |
| --- | --- |
| Optimizer | Adam [42] |
| $\beta_1$ | 0.9 |
| $\beta_2$ | 0.999 |
| Learning Rate | $2 \times 10^{-4}$ |
| $\lambda$ (Equation (5)) | 10 |
| Training Batch Size | 2 |
| Testing Batch Size | 1 |

In the ablation study, we use a slightly different baseline model in order to prove the effectiveness of perceptual loss. The only difference is that the perceptual loss is disabled at first and then enabled in the ablation study. Below, we show how this baseline model evolves step by step to become our final model and what role each key technique plays in our model.

### 4.3. Evaluation Schemes and Metrics

We perform quantitative analysis using the following performance indices:

- Fréchet Inception Distance (FID) [43]: It is a refinement of the inception score [44] and compares the mean and covariance of an Inception-v3 [45] network's (pre-trained on ImageNet [46]) intermediate features for real and synthesized images. We employ FID for both datasets. Lower FID values mean closer distances between synthetic and real data distributions. The lower bound of FID is 0.
- Peak Signal to Noise Ratio (PSNR) [47]: This is a paired image quality assessment. It is a commonly used pixel-by-pixel measurement. Greater PSNR values indicate better quality.
- Structural Similarity Index Measures (SSIM) [48]: As another paired image quality assessment, the SSIM measurement is closer to human perception compared to PSNR [49] because it considers the inter-dependencies between pixels within a window of a specific size. We additionally adopt this metric as a measurement of the similarity between a synthesized image and the corresponding target image. To calculate the SSIM, we set the window's size to 11. The upper bound of the SSIM is 1.0. A score closer to 1.0 indicates better quality.
- Precision-Recall [50]: Precision and recall metrics are proposed as an alternative to FID when assessing the performance of GANs [50,51]. Precision quantifies the similarity of generated samples to the real ones, while recall denotes the capacity of a generator to produce all instances present in the set of real images (Figure 9). These metrics aim to explicitly quantify the trade-off between diversity (recall) and quality (precision).

### 4.4. Comparisons of Simple Scenario Datasets

We compare our method to Pix2Pix and the baseline model using the simple scenario dataset. The corresponding results are presented in Figure 10. We show the generation output and enlarged regional details for each method in adjacent rows. The first two columns of Figure 10 show the real optical images of $T_0$ and $T_1$. As the remote sensing images were acquired between summer and spring, we can see that the vegetation has changed its color from $T_0$ to $T_1$. From the last three columns of Figure 10, we can see that all three methods evaluated are capable of capturing this variety. Generally speaking, all evaluated methods show adequate correspondence between real and generated images under the simple scenario.

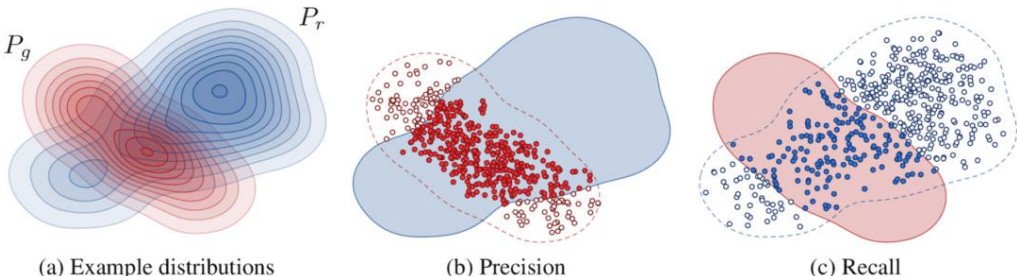

(a) Example distributions                (b) Precision                (c) Recall

**Figure 9.** Illustration of precision and recall for distributions, reprinted with permission from Ref. [50], 2019, Kynkäänniemi et al. (**a**) The distribution of real images ($P_r$) and the distribution of generated images ($P_g$). (**b**) Precision is defined as the probability of a randomly selected image from $P_g$ being encompassed by the scope of $P_r$. (**c**) Recall is defined as the probability of a randomly selected image from $P_r$ being encompassed by the scope of $P_g$.

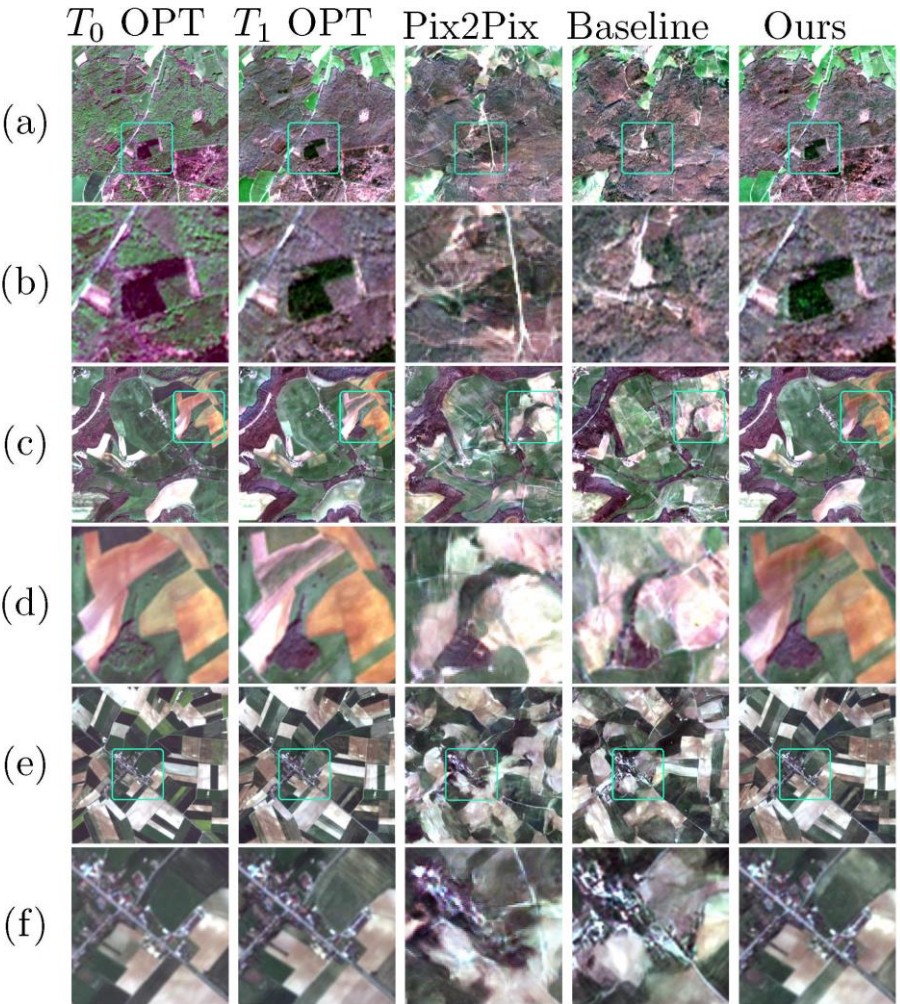

**Figure 10.** Visual comparison under a simple scenario. Three samples are given in (**a**,**c**,**e**). Some regions in (**a**,**c**,**e**) are marked with rectangles and enlarged in (**b**,**d**,**f**) respectively.

When closely zoomed in, the images synthesized by our method can be hardly distinguished from the real ones. As shown in Figure 10b,f, the details, such as the contours of small ponds and country paths, are better restored by our method than by other methods. Additionally, Figure 10d shows that our method better restores the color of the fields. The output of the baseline model presents fewer realistic details, but it still outperforms Pix2Pix.

In addition to visual inspection, the quantitative analysis also proves better performance by our model. Table 2 reports the evaluation metrics for these methods. Bold indicates optimal performance. As can be seen, the proposed method outperforms the other two methods on all metrics.

**Table 2.** Evaluation metrics comparison under a simple scenario.

| Metrics | Pix2Pix | Baseline | Ours |
|---------|---------|----------|------|
| FID $\downarrow$ | 221.1 | 99.9 | **69.46** |
| PSNR $\uparrow$ | 13.80 | 18.36 | **21.89** |
| SSIM $\uparrow$ | 0.23 | 0.62 | **0.78** |

Our method achieves the best score on all metrics, especially on FID, with more than 30% improvement compared to the baseline model.

### 4.5. Comparisons of Complicated Scenario Datasets

Under a simple scenario, both the baseline model and our method have visually acceptable results. Under a complicated scenario, however, the results of different methods can vary greatly because the challenge of various kinds of small-scale objects is a tricky issue faced by every method. So, it is necessary to carefully evaluate our method and compare it with a wider range of other methods. We conducted a quantitative and qualitative comparison with a number of advanced supervised methods, including the baseline model, PSP [22], Selection-GAN [26], CHAN [25], and VQGAN transformer [52], the last three of which are good competitors because they also leverage the power of an attention mechanism. We input the last four methods with the same input bands as our method, while the baseline model is still input with bi-temporal SAR imagery.

Figure 11 shows the optical images synthesized by different methods. The regions where changes have occurred between $T_0$ and $T_1$ are marked with rectangles and enlarged for detailed comparison. Visual inspection reveals that PSP even fails to generate globally fine output, seriously affecting the interpretation of land types. VQGAN transformer has insufficient learning ability for changes, resulting in the generated images being very insensitive to regional changes and looking similar to $T_0$ optical images. The other three methods show some ability to alleviate these phenomena. However, they also show some shortages in terms of preserving colors and detail learning ability. For example, CHAN tends to have obvious color distortion with a very bright color in some areas, especially changing areas. This phenomenon has also seriously affected the presentation of details in these areas. The detail learning ability of the baseline model has totally failed under a complicated scenario. Although its general performance is better than PSP's, the local details are seriously deformed, with clearly observable geometric errors in object boundaries. As for Selection-GAN, it is a very competitive method and would yield the best output if our method were not taken into consideration. However, it makes blur inferences within changing areas, generating heavily smoothed objects. Sometimes it is acceptable to make blur inferences on changes, e.g., vegetation's natural decline; however, many artificial changes, such as constructing buildings and opening new roads, require inferences of rich texture and sharp contours.

In contrast, the optical features, change features, and multi-scale features are fully extracted by the method we propose, so that the details and change information are well-learned. As shown in the last column of Figure 11, the changes in vegetation decline (Figure 11f) and building construction (Figure 11b,d) can all be recognized in the synthesized images with rich details and clear boundaries. As shown in Figure 12, we randomly selected some pixels from the generated images, finding that their values are in good correspondence with those selected from the real images. The quality of synthesized images is also presented with evaluation metrics in Table 3. It is notable that under a complicated scenario, our model outperforms the baseline model by a large margin. Additionally, it is also very inspiring to find that our method gets the best score on most evaluation metrics. Among all the evaluated methods, the scores of Selection-GAN are as good as

our method, even better on the SSIM, which is not beyond our expectation because the model has yielded very competitive results in visual inspections. The success of Selection-GAN and our method further proves the important role attention mechanisms can play in generative models.

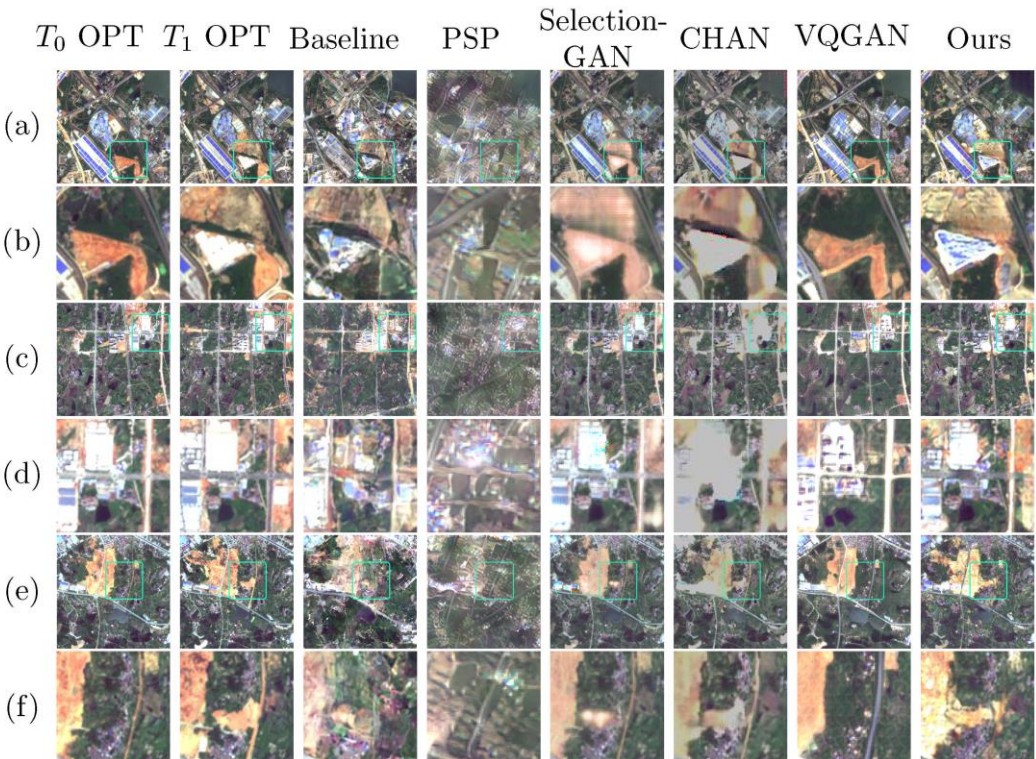

**Figure 11.** Visual comparison under a complicated scenario. Three samples are given in (**a**,**c**,**e**). Some regions in (**a**,**c**,**e**) are marked with rectangles and enlarged in (**b**,**d**,**f**) respectively.

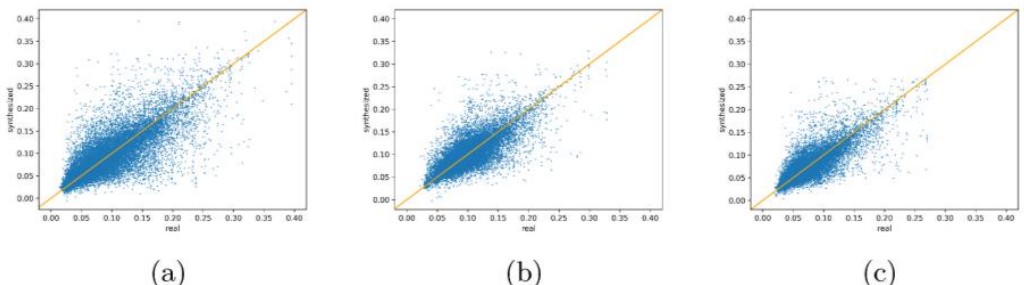

**Figure 12.** Scatter plot of generated and real pixel values. (**a**) a red band; (**b**) a green band; (**c**) a blue band.

**Table 3.** Evaluation metrics comparison under a complicated scenario.

| Metrics | Baseline | PSP | Selection-GAN | CHAN | VQGAN Transformer | Ours |
|---|---|---|---|---|---|---|
| FID ↓ | 121.35 | 148.10 | 82.02 | 112.36 | 108.14 | **80.60** |
| PSNR ↑ | 12.14 | 14.0 | 18.53 | 17.55 | 13.63 | **20.79** |
| SSIM ↑ | 0.17 | 0.19 | **0.66** | 0.60 | 0.25 | 0.62 |

Our method ranks first on FID and PSNR and second on SSIM, which is 6.06% lower than Selection-GAN.

We also compare the generalization performance of our model with that of the baseline model by observing how the precision and recall metrics change when gradually adding new samples. The precision and recall represent the models' ability to generate effective results and to generate diverse results, respectively. We first perform the inference with

some learned samples, which results in a high precision and a rather low recall. Then we gradually add unlearned samples from other locations and times to the testing set and perform inference on these samples. Because these samples are pretty different from those that have been learned by the models, generating effective results can be harder, but the diversity of the results will increase. So even though higher values for both precision and recall are preferred, there will be an inevitable trade-off between the two metrics. Figure 13 presents the precision-recall (PR) curves for both methods. With recall increasing, the precision of both methods is steady at first but then drops severely after some point. However, compared to the baseline model, the precision drop of our method comes much slower, which shows that our method is better at keeping a high level of both precision and recall when generalizing to new datasets.

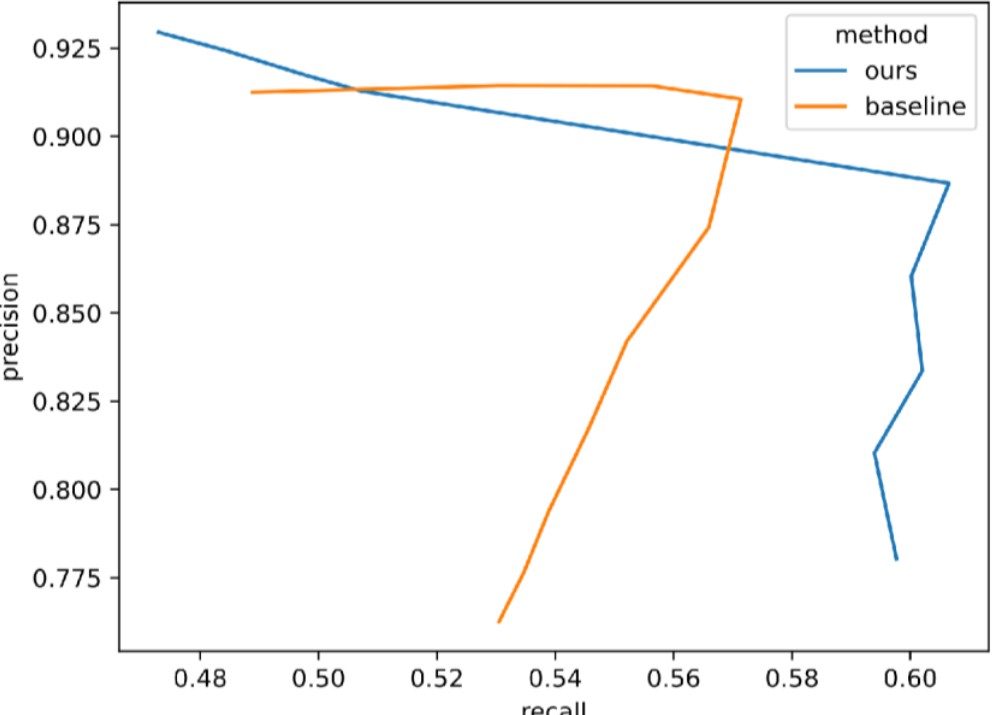

**Figure 13.** The PR curve comparison.

*4.6. Ablation Study*

This study intends to assess every component of the proposed method. All experiments have been conducted on our complicated scenario dataset to show how these key techniques improve the model's performance under a complicated scenario. We disable the perceptual loss in the baseline model used in the comparison experiments to form a different baseline model named B. Then we sequentially add multi-scale generation (+l), perceptual loss (+pl), additional optical input (+opt), weight-balancing strategy (+wb), and temporal co-attention (+at) to form the proposed model. The results are reported in Figure 14 and Table 4. We can see significant improvements with additional optical input in the first six columns of Figure 14, while the improvements with multi-scale generation, weight-balancing strategy, and temporal co-attention are less notable through visual inspection. Nevertheless, the evaluation metrics in Table 4 show that all key techniques have brought improvements in the evaluation indicators to some extent, some of which are rather significant while others are milder. The technique with the greatest contribution to performance is adding additional optical input. It brings at least 16% improvements in all evaluation indicators. For the SSIM indicator, it even doubles the result of B + l + pl. The second biggest contribution comes from the temporal co-attention mechanism, with about a 2% improvement in FID and at least 23% improvements in other evaluation indicators.

The third biggest contribution comes from perceptual loss, with about a 17% improvement in FID and SSIM and a slight improvement in PSNR.

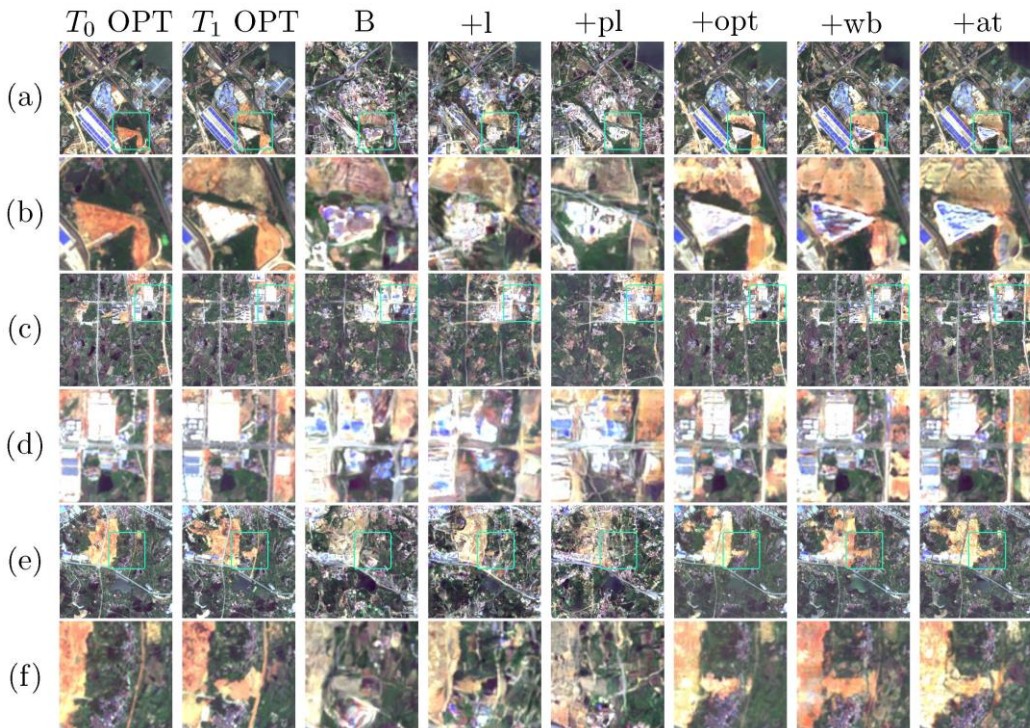

**Figure 14.** Visual inspection of the ablation study. Columns three to eight show the results of B, B + l, B + l + pl, B + l + pl + opt, B + l + pl + opt + wb, and B + l + pl + opt + wb + at, respectively. Three samples are given in (**a**,**c**,**e**). Some regions in (**a**,**c**,**e**) are marked with rectangles and enlarged in (**b**,**d**,**f**) respectively.

**Table 4.** Evaluation metrics results of the ablation study.

| Metrics | B | B + l | B + l + pl | B + l + pl + opt | B + l + pl + opt + wb | B + l+pl + opt + wb + at |
|---|---|---|---|---|---|---|
| FID ↓ | 149.03 | 133.80 (−10.22%) * | 108.25 (−27.36%) | 84.43 (−43.35%) | 83.53 (−43.95%) | **80.60** (−45.92%) |
| PSNR ↑ | 11.97 | 12.34 (+3.09%) | 13.33 (+11.36%) | 16.45 (+37.43%) | 17.96 (+50.04%) | **20.79** (+73.68%) |
| SSIM ↑ | 0.17 | 0.18 (+5.88%) | 0.21 (+23.53%) | 0.58 (+241.18%) | 0.57 (+235.29%) | **0.62** (+264.71%) |

\* The percentages in brackets are the improvement compared to the baseline model.

## 5. Conclusions

In this article, a new GAN-based SAR-to-optical image translation method is proposed to improve the quality of synthesized optical images, especially in a complicated scenario. The results in this paper suggest that using additional optical input and a temporal co-attention-guided generator can greatly boost the performance of a CGAN. We have observed that incorporating multi-scale generation and a weight-balancing strategy can also help yield better results.

Additionally, we qualitatively and quantitatively compared our method with a number of advanced supervised methods on two new datasets that we built. The results demonstrated the superiority of the proposed method in both simple and complicated scenarios. We also found that, among all evaluated models, leveraging attention mechanisms helped the models yield competitive results. This indicates a possibility for using various kinds of attention mechanisms together to advance the further progression of SAR-to-optical image translation. We will conduct such studies in the future. In addition, there are some

factors that can affect the model's performance that need further exploration. For instance, considering that involving SAR-to-optical translation with multi-temporal methodology is a little-studied direction, how the time interval of our bi-temporal dataset can affect the results needs to be carefully studied in future works.

**Author Contributions:** Data curation, J.Y.; Investigation, Y.W.; Methodology, Y.W. and Y.M.; Project administration, Y.M., F.C. and J.L.; Resources, F.C. and S.Z.; Supervision, S.Z. and J.Y.; Validation, Y.W., E.S. and W.Y.; Writing—original draft, Y.W.; Writing—review & editing, Y.W. and Y.M. All authors have read and agreed to the published version of the manuscript.

**Funding:** This work was funded by the National Natural Science Foundation of China (grant number 42201063), the Key Research and Development Program of Hainan Province (grant number ZDYF2021SHFZ260) and Hainan Provincial Natural Science Foundation of China (grant number 322QN345, 520QN295). We thank for their support.

**Data Availability Statement:** The data of experimental images used to support the findings of this. research is available from the corresponding author upon reasonable request.

**Acknowledgments:** We thank Isola et al. [10], Wang et al. [23], Karras et al. [21], Tang et al. [26], Gao et al. [25] and Esser et al. [52] for sharing their code. We also thank ESA for providing the primary Sentinel-1 and Sentinel-2 data.

**Conflicts of Interest:** The authors declare no conflict of interest.

## Abbreviations

| | |
|---|---|
| GANs | Generative adversarial networks |
| CGAN | Conditional generative adversarial network |
| SAR | Synthetic aperture radar |
| I2I | Image-to-image |
| BOA | Bottom-of-Atmosphere |
| LSGANs | Least squares generative adversarial networks |
| PSNR | Peak signal-to-noise Ratio |
| SSIM | structural similarity index measure |
| FID | Fréchet Inception Distance |
| PR | precision-recall |

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
