# Peer review of "Temporal Co-Attention Guided Conditional Generative Adversarial Network for Optical Image Synthesis"

_remotesensing, doi:10.3390/rs15071863_

Round 1

Reviewer 1 Report

Authors provided an interesting paper which is clearly of interest for Remote Sensing journal readers. However, to reach the necessary quality to be accepted,as far as I am concerned, the manuscript needs to be improved. 

One aspect that I consider necessary is observing the literature review. I noticed the absence of versions published in scientific journals published by the authors of articles in conferences which you cited in that bibliographic review. I think it's worth including them in the references. I quote some articles published by authors whom I know personally: 

J. Noa Turnes, J. D. B. Castro, D. L. Torres, P. J. S. Vega, R. Q. Feitosa and P. N. Happ, "Atrous cGAN for SAR to Optical Image Translation," in IEEE Geoscience and Remote Sensing Letters, vol. 19, pp. 1-5, 2022, Art no. 4003905, doi: 10.1109/LGRS.2020.3031199.

J. D. Bermudez, P. N. Happ, R. Q. Feitosa and D. A. B. Oliveira, "Synthesis of Multispectral Optical Images From SAR/Optical Multitemporal Data Using Conditional Generative Adversarial Networks," in IEEE Geoscience and Remote Sensing Letters, vol. 16, no. 8, pp. 1220-1224, Aug. 2019, doi: 10.1109/LGRS.2019.2894734.

Thus, I consider to be necessary to look into this aspect in a broader manner, since methods and results presented in articles published in journals tend to have been better matured than their preliminary versions presented in conferences.

Another issue that I consider necessary to revisit corresponds to the presentation of the method. This one, in my opinion, needs a good revision focusing on improving the clarity of its presentation. I think that despite my experience with generative methods and cGAS, I had a bad experience reading section 3.

To end up, I would like to mention that I very much appreciated the design and presentation of the experiments.

Reviewer 2 Report

In this manuscript, a novel conditional generative adversarial network method with temporal co-attention mechanism is proposed. By utilizing the correlation between optical-available and optical-absent time points and a weight-balancing strategy on SAR and optical data, the proposed method can achieve high-quality SAR-to-optical image translation performance under complicated scenario. The organization of the manuscript is relatively fair and the simulation results are reasonable. However, there are still some defects need to be improved. The detailed comments are listed below.

1. In this paper, you can briefly explain each part of Figure 5, or make a clearer expression in Figure 5. For example, as shown in the sub-figure ‘Input Patches’ in Figure 5, the meaning of 7 here is the number of channels, but the bracket includes five images, which is easy create ambiguity.

2. In the loss function expressions of Eq. 2 and Eq. 3, the generator is represented by G(s), which does not reflect the two generators used in the multi-scale generation mentioned above.

3. This manuscript adds perceptual loss to the loss function of the generator, and ablation experiments should be added to prove the effectiveness of perceptual loss.

4. Detailed descriptions of the conditional GAN and transformer should be given in the introduction or related work.

5. In the introduction section, some important references on the application of deep learning in SAR image processing are omitted, such as:

1)    https://www.doi.org/10.1109/LGRS.2022.3177001

2)    https://www.doi.org/10.1109/TGRS.2023.3248040

3)    https://www.doi.org/10.1109/JSTARS.2022.3199091

Round 2

Reviewer 2 Report

The authors have revised the manuscript according to the comments of reviewers. I have no other questions.